# Clinicopathological characteristics and prognosis analysis of gastrinoma based on the SEER database

**Chenjian Qiu[1]**, **Chengru Hu[2]**, **Chengjie Lu[3]**, **Jingxian Shi[3]**, **Zhenguo Qiao[4]***,
**Yaping Wang[5]***, **Feidie Xu[3]***

**1** Department of Rehabilitation, Suzhou Ninth People's Hospital, The Affiliated Suzhou Bay Clinical College of Xuzhou Medical University, Suzhou, China, **2** Department of Oncology, Suzhou Ninth People's Hospital, The Affiliated Suzhou Bay Clinical College of Xuzhou Medical University, Suzhou, China, **3** Department of Traditional Chinese Medicine, Suzhou Ninth People's Hospital, The Affiliated Suzhou Bay Clinical College of Xuzhou Medical University, Suzhou, China, **4** Department of Gastroenterology, Suzhou Ninth People's Hospital, The Affiliated Suzhou Bay Clinical College of Xuzhou Medical University, Suzhou, China, **5** Department of Gastroenterology, Affiliated Huishan Hospital of Xinglin College, Nantong University, Wuxi Huishan District People's Hospital, Wuxi, China

☺ These authors contributed equally to this work.
* xfd525@163.com (FX); qzg66666666@163.com (ZQ); rose19820721@126.com (YW)

## Abstract

Gastrinoma, distinguished by Zollinger-Ellison Syndrome, is a highly unusual neoplasm. However, understanding of its clinicopathological characteristics and survival at the population level is inadequate. This study aimed to examine the clinicopathological features and survival of gastrinoma patients utilizing the Surveillance, Epidemiology, and End Results (SEER) database. Patients diagnosed with gastrinoma from 2000 to 2020 were included in the study. Multiple imputation was used to handle missing data. Chi-square test was employed to analyze the clinicopathological features. The Kaplan-Meier method and Cox proportional hazards models were utilized to evaluate overall survival (OS) and cancer-specific survival (CSS). A total of 160 patients participated in the study. The incidence rate rose with age, peaking at 60–69 years. Notably, the most common site was the pancreas (52.5%), followed by the duodenum (32.5%) and stomach (10.6%). Most gastrinomas were well-differentiated (77.5%). Liver was a frequent metastasis site, with age and tumor size identified as risk factors by multivariate analysis. The 1-, 3-, 5-, and 10-year OS rates were 92.4%, 84.3%, 77.6%, and 62.9%, respectively, with corresponding CSS rates of 94.8%, 89.2%, 86.0%, and 75.2%. Additionally, duodenal gastrinomas showed better differentiation, earlier staging, smaller size, and fewer metastases than pancreatic ones (P < 0.05). Pancreatic gastrinomas had poorer OS (hazard ratio [HR] 2.38, 95% confidence interval [CI] 1.30–4.35, P = 0.005) and CSS (HR 10.70, 95% CI 2.55–44.90, P = 0.001) compared to duodenal ones. Multivariate Cox analysis highlighted N stage, surgery, chemotherapy, and tumor size as risk factors for both OS and CSS.

**Data availability statement:** All relevant data are within the paper and its Supporting Information files.

**Funding:** This work was supported by the Science and Technology Development Program of Suzhou (SYWD2024077), the Development Fund of Xuzhou Medical University Affiliated Hospital (XYFZ202404 and XYFM202426), the Program for the Talents in Science and Education of Wujiang District, Suzhou, China (Grant No. WWK202117) and the Scientific Research Project of Suzhou Ninth People's Hospital (YK202117 and YK202438). There was no additional external funding received for this study. The funders had no role in study design, data collection and analysis, decision to publish, or preparation of the manuscript.

**Competing interests:** The authors have declared that no competing interests exist.

Gastrinomas located in the duodenum and pancreas exhibit distinct clinicopathological characteristics. The prognosis for gastrinomas in the duodenum is better than that for gastrinomas in the pancreas.

## Introduction

Neuroendocrine neoplasms (NENs) represent a heterogeneous group of tumors that originate from peptidergic neurons and neuroendocrine cells [1,2]. These tumors are classified as either functional (F-NENs) or non-functional based on whether they secrete hormones and cause corresponding symptoms. F-NENs account for approximately 20% of all NENs [3,4]. Functional neuroendocrine tumors can be further divided into subtypes according to their primary anatomical location and the hormone they produce. These subtypes include functional gastrointestinal neuroendocrine tumors (F-GINENs), functional pancreatic neuroendocrine tumors (F-pNENs), functional bronchopulmonary neuroendocrine tumors, and functional thymic neuroendocrine tumors [5–8]. Gastrinoma, a specific subtype of F-GINEN, frequently manifests as Zollinger-Ellison Syndrome (ZES). Patients with ZES typically present with a range of symptoms, such as abdominal pain, intermittent diarrhea, persistent or atypical peptic ulcers, and gastroesophageal reflux. These symptoms usually improve with the use of proton pump inhibitors but tend to recur once the medication is discontinued [9]. A notably elevated level of fasting serum gastrin serves as a crucial diagnostic marker for gastrinoma [10]. Although the majority of gastrinomas occur sporadically, a substantial proportion is linked to multiple endocrine neoplasia type 1 (MEN1). MEN1 is an autosomal dominant disorder characterized by the presence of parathyroid tumors, anterior pituitary adenomas, and pNENs [11,12]. At present, surgical intervention stands as the mainstay of treatment for gastrinoma [13,14]. For patients who are not candidates for surgery or experience disease recurrence after surgery, medications like proton pump inhibitors, H2 receptor blockers, and somatostatin analogs are prescribed to alleviate symptoms. In cases of malignant gastrinoma or when metastasis has occurred, chemotherapy may be utilized to extend survival [15–17].

Previous studies have provided insights into the clinicopathological features and prognosis of gastrinoma. For instance, Cavalcanti et al. [18] detailed the clinicopathological attributes of a 24-year-old male patient with a lymph node gastrinoma, while Chen et al. [19] presented the clinical traits and outcomes of 39 surgically treated gastrinoma patients, finding an average age of onset at 53 years, a balanced gender distribution, and 5-year and 10-year overall survival (OS) rates of 80.8% and 60.7% respectively, with approximately one-fourth of the cases being primary lymph node gastrinomas. Alexander et al. [20] evaluated the utility of somatostatin receptor scintigraphy in locating gastrinoma in 35 Zollinger-Ellison Syndrome (ZES) patients, identifying 74 gastrinomas including 22 in the duodenum, 8 in the pancreas, 3 in other sites, and 41 as lymph node metastases. Mortellaro et al. [21] reported favorable long-term outcomes following local gastrinoma removal in 12 MEN-1 patients with ZES who underwent surgical management. However, existing reports

on gastrinoma's clinical manifestations are limited by small sample sizes, predominantly consisting of case reports, and a lack of comprehensive long-term follow-up data on prognosis. To overcome these limitations and gain a more thorough understanding of gastrinoma, we conducted an analysis using data from the Surveillance, Epidemiology, and End Results (SEER) database at a population level, aiming to enhance our understanding of gastrinoma's clinicopathological characteristics and prognosis and thereby inform clinical management.

## Materials and methods

### Patient selection

This study centered on patients diagnosed with gastrinoma between 2000 and 2020, using data sourced from the SEER database. Patient identification was achieved by employing the ICD-O-3 (International Classification of Diseases for Oncology, Third Edition), with the specific code 8153/3 designating malignant gastrinoma. Patients with unknown survival information were excluded from this study. A wide range of factors were examined, including marital status, age, race, gender, tumor location, tumor grade, TNM stage, tumor size, whether surgical intervention was performed, chemotherapy status, liver metastasis, lung metastasis, overall survival (OS), cancer-specific survival (CSS), and the follow-up period. Age was divided into 10-year intervals: < 29, 30–39, 40–49, 50–59, 60–69, and >69 years. Fig 1 illustrates the age distribution and the number of cases, revealing a peak in incidence after the age of 60. For statistical analysis, patients were split into a younger group (<60 years) and an older group (≥60 years). Race was classified as white, black, or other races, which included American Indian/Alaska Native and Asian/Pacific Islander ethnicities. Marital status was categorized as either married or unmarried, with the unmarried category covering divorced, separated, single, or widowed individuals. Tumor size was grouped into ≤2.0, 2.1–5.0, and >5.0 cm to reflect clinical stages, the risk of metastasis, and prognostic differences. Tumor locations with fewer than 10 samples were grouped under "Others," mainly consisting of 5 cases in the small intestine, 1 case in the cecum, and 1 case in the ampulla of Vater. The objective of the study was to assess the clinicopathological characteristics and prognosis of gastrinoma, as well as to compare the differences in clinicopathological features and prognosis between gastrinomas located in the duodenum and those in the pancreas. Both OS and CSS

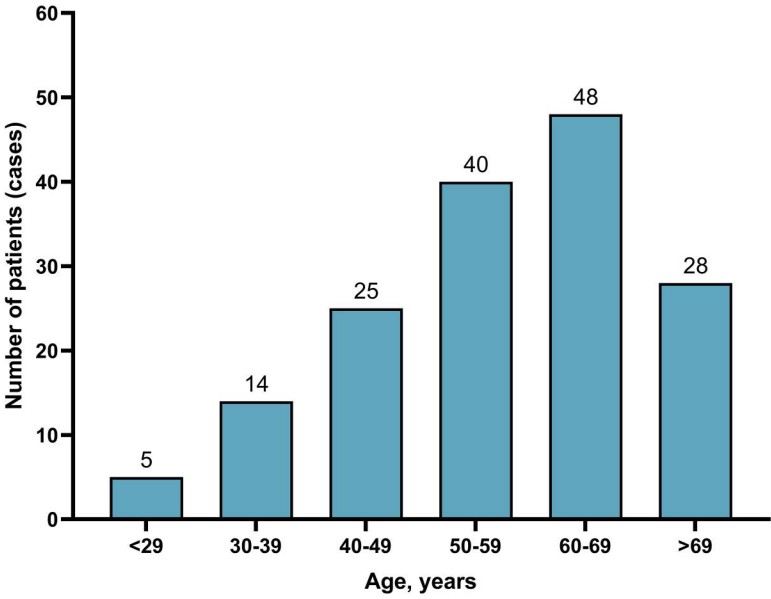

**Fig 1. The age distribution of patients with gastrinoma.**

were calculated from the date of gastrinoma diagnosis until death, cancer-related death, or the end of the follow-up period, whichever came first. Since the SEER database provides publicly accessible anonymized data, neither IRB (Institutional Review Board) approval nor explicit consent was necessary.

## Statistical analysis

Categorical variables were expressed as frequencies and percentages, and Chi-square tests were employed to evaluate differences among groups. For continuous data that did not follow a normal distribution, the median and interquartile range (IQR) were reported, and comparisons were made using the Mann – Whitney U test. To address missing data in the SEER database, we applied multiple imputation (MI) using the mice package (version 4.1.0) in R. Specifically, we generated 20 imputed datasets by utilizing logistic regression for categorical variables. We assessed convergence both visually through trace plots and statistically using the Gelman-Rubin diagnostic. These assessments confirmed that the imputation models had achieved convergence. The pooled results from these 20 imputed datasets were then used in subsequent analyses to enhance robustness and reduce bias resulting from missing data. For the survival analysis of both OS and CSS, we applied the Kaplan-Meier method. Survival curves were compared using the log-rank test. Additionally, Cox proportional hazards models were used for both univariate and multivariate analyses to calculate hazard ratios (HRs) along with their corresponding 95% confidence intervals (CIs). Variables with a P-value less than 0.05 in the univariate analysis were included in the multivariate Cox regression model. A univariate analysis was carried out to identify potential risk factors associated with liver metastasis. Factors with a P-value below 0.05 from this univariate analysis were subsequently analyzed using a multivariate logistic regression model. All statistical calculations were performed using R software (version 4.1.0). Statistical significance was defined as a P-value less than 0.05.

## Results

### Patients characteristics

A study encompassing 160 patients found that the prevalence of gastrinoma increased with age, reaching its peak among those aged 60–69. In terms of demographic characteristics, males accounted for 48.1% of the patients, while married individuals made up 52.5%, and Caucasians were the predominant ethnic group, representing 77.5% of the cohort. Regarding the anatomical location of gastrinomas, the pancreas was the most common site, with 52.5% of cases originating there, followed by the duodenum (32.5%) and the stomach (10.6%). The majority of gastrinomas were well-differentiated, comprising 77.5% of the total. In terms of metastasis, lymph node involvement was observed in 45.0% of cases, and distant metastasis occurred in 21.9%, with the liver being a frequent site of spread, seen in 18.1% of instances, while lung metastasis was rare, occurring in only 1.3% of cases. Table 1 provides a comprehensive summary of the clinicopathological features of gastrinoma patients. Additionally, S1 Table offers a detailed overview of the clinicopathological features of patients before multiple imputation (MI), highlighting that marital status data were missing in 5.6% of cases, grade information in 61.9%, TNM stage details in 54.4%, surgery records in 1.3%, tumor size data in 45.0%, and data on both liver and lung metastasis in 55.0% of cases.

### Survival analysis of patients with gastrinoma

The gastrinoma patients demonstrated 1-, 3-, 5-, and 10-year OS rates of 92.4%, 84.3%, 77.6%, and 62.9% respectively, along with corresponding CSS rates of 94.8%, 89.2%, 86.0%, and 75.2%. Following surgery, these patients showed improved OS rates at the same time intervals, namely 94.8%, 90.3%, 85.5%, and 74.5%, with CSS rates of 96.8%, 94.5%, 93.2%, and 87.3% respectively. When considering the anatomical location of gastrinomas, patients with tumors in the duodenum had 1-, 3-, 5-, and 10-year OS rates of 96.1%, 93.8%, 93.8%, and 79.9%, with matching CSS rates of 98.0%, 98.0%, 98.0%, and 94.5%. In contrast, those with pancreatic gastrinomas exhibited lower OS rates at these time

**Table 1. Baseline characteristics of patients with gastrinoma based on the SEER database.**

| Variables | N = 160 | % |
|---|---|---|
| Marital status | | |
| Married | 84 | 52.5 |
| Unmarried | 76 | 47.5 |
| Age, years | | |
| <29 | 5 | 3.1 |
| 30-39 | 14 | 8.8 |
| 40-49 | 25 | 15.6 |
| 50-59 | 40 | 25.0 |
| 60-69 | 48 | 30.0 |
| >69 | 28 | 17.5 |
| Age, years, median (IQR) | 58.0 (47.3, 65.8) | |
| Race | | |
| White | 124 | 77.5 |
| Black | 27 | 16.9 |
| Others | 9 | 5.6 |
| Location | | |
| Stomach | 17 | 10.6 |
| Duodenum | 52 | 32.5 |
| Pancreas | 84 | 52.5 |
| Others | 7 | 4.4 |
| Grade | | |
| Well | 124 | 77.5 |
| Moderately | 22 | 13.8 |
| Poorly | 14 | 8.8 |
| Sex | | |
| Male | 77 | 48.1 |
| Female | 83 | 51.9 |
| T stage | | |
| T1 | 31 | 19.4 |
| T2 | 47 | 29.4 |
| T3 | 15 | 9.4 |
| T4 | 29 | 18.1 |
| TX | 38 | 23.8 |
| N stage | | |
| N0 | 68 | 42.5 |
| N1 | 72 | 45.0 |
| NX | 20 | 12.5 |
| M stage | | |
| M0 | 125 | 78.1 |
| M1 | 35 | 21.9 |
| Surgery | | |
| Yes | 98 | 61.3 |
| No | 62 | 38.8 |
| Chemotherapy | | |
| Yes | 13 | 8.1 |
| No/Unknown | 147 | 91.9 |

*(Continued)*

**Table 1.** (Continued)

| Variables | N = 160 | % |
|---|---|---|
| Tumor size, cm | | |
| ≤2.0 | 76 | 47.5 |
| 2.1-4.0 | 37 | 23.1 |
| >4.0 | 47 | 29.3 |
| Liver metastasis | | |
| No | 131 | 81.9 |
| Yes | 29 | 18.1 |
| Lung metastasis | | |
| No | 158 | 98.8 |
| Yes | 2 | 1.3 |
| Survival months, median (IQR) | 82.0 (34.3, 146.0) | |

Seer: surveillance, epidemiology, and end results; IQR: interquartile range

points, being 90.5%, 79.2%, 68.5%, and 48.7% respectively, along with CSS rates of 93.9%, 84.8%, 78.8%, and 60.0% (as detailed in Table 2). Additionally, Fig 2 presents the OS and CSS trends for all gastrinoma patients, while Fig 3 specifically illustrates the OS and CSS of gastrinoma patients post-surgery.

## Comparison of clinicopathological features and prognosis between gastrinomas located in the duodenum and pancreas

Upon comparison between duodenal and pancreatic gastrinomas, it became evident that duodenal gastrinomas exhibited better differentiation, were diagnosed at an earlier stage, had smaller tumor sizes, and presented with fewer metastases, with all these differences reaching statistical significance (P < 0.05), as elaborated in Table 3. This table also offers a thorough summary of the clinicopathological characteristics of gastrinomas originating from both the duodenum and the pancreas. Notably, when assessing survival outcomes, pancreatic gastrinomas were associated with a significantly poorer OS, as indicated by a HR of 2.38 (95% CI: 1.30–4.35, P = 0.005) (Fig 4), and a markedly worse CSS, with an HR of 10.70 (95% CI: 2.55–44.90, P = 0.001) (Fig 5), compared to their duodenal counterparts.

## Univariate and multivariate cox regression

In the univariate Cox regression analysis, we identified that age, tumor grade, N stage, M stage, surgical intervention, chemotherapy treatment, tumor dimensions, and the presence of liver metastasis were all significant risk factors influencing the OS of patients with gastrinoma. Subsequent multivariate Cox regression analysis further refined these findings, revealing that age, N stage, surgical procedures, chemotherapy, and tumor size independently impacted gastrinoma OS, as detailed in Table 4. Similarly, univariate Cox regression analysis indicated that race, tumor grade, T stage, N stage, M stage, surgical treatment, chemotherapy administration, tumor size, and liver metastasis were all factors associated with CSS in gastrinoma patients. Upon conducting multivariate Cox regression analysis, we established that tumor grade, N stage, surgical interventions, chemotherapy, and tumor size were independent risk factors for gastrinoma CSS, as presented in Table 5.

## Univariate and multivariate analysis of liver metastasis

A univariate analysis examining risk factors among 29 patients with liver metastasis revealed that, when comparing the liver metastasis group to the non-liver metastasis group, there were no significant differences in marital status, race, tumor

**Table 2. Overall survival and cancer-specific survival analysis in patients with gastrinoma.**

| | Survival (%) | 95%CI Lower | 95%CI Upper |
|---|---|---|---|
| Overall survival | | | |
| Gastrinoma (all) | | | |
| 1-year | 92.38 | 88.32 | 96.62 |
| 3-year | 84.26 | 78.66 | 90.26 |
| 5-year | 77.61 | 71.12 | 84.69 |
| 10-year | 62.87 | 55.00 | 71.87 |
| Gastrinoma (all) after surgery | | | |
| 1-year | 94.78 | 90.42 | 99.34 |
| 3-year | 90.30 | 84.47 | 96.54 |
| 5-year | 85.51 | 78.49 | 93.15 |
| 10-year | 74.50 | 65.50 | 84.75 |
| Gastrinoma located in duodenum | | | |
| 1-year | 96.08 | 90.90 | 100.00 |
| 3-year | 93.84 | 87.31 | 100.00 |
| 5-year | 93.84 | 87.31 | 100.00 |
| 10-year | 79.89 | 68.09 | 93.73 |
| Gastrinoma located in pancreas | | | |
| 1-year | 90.45 | 84.36 | 96.97 |
| 3-year | 79.18 | 70.82 | 88.51 |
| 5-year | 68.49 | 58.94 | 79.59 |
| 10-year | 48.66 | 37.97 | 62.35 |
| Cancer-specific survival | | | |
| Gastrinoma (all) | | | |
| 1-year | 94.79 | 91.33 | 98.37 |
| 3-year | 89.20 | 84.32 | 94.35 |
| 5-year | 86.00 | 80.46 | 91.92 |
| 10-year | 75.21 | 67.78 | 83.46 |
| Gastrinoma (all) after surgery | | | |
| 1-year | 96.79 | 93.29 | 100.00 |
| 3-year | 94.45 | 89.82 | 99.31 |
| 5-year | 93.24 | 88.14 | 98.63 |
| 10-year | 87.27 | 80.07 | 95.12 |
| Gastrinoma located in duodenum | | | |
| 1-year | 98.04 | 94.31 | 100 |
| 3-year | 98.04 | 94.31 | 100 |
| 5-year | 98.04 | 94.31 | 100 |
| 10-year | 94.54 | 87.17 | 100 |
| Gastrinoma located in pancreas | | | |
| 1-year | 93.85 | 88.77 | 99.22 |
| 3-year | 84.77 | 77.18 | 93.10 |
| 5-year | 78.83 | 70.06 | 88.69 |
| 10-year | 60.00 | 48.71 | 73.91 |

CI: confidence interval

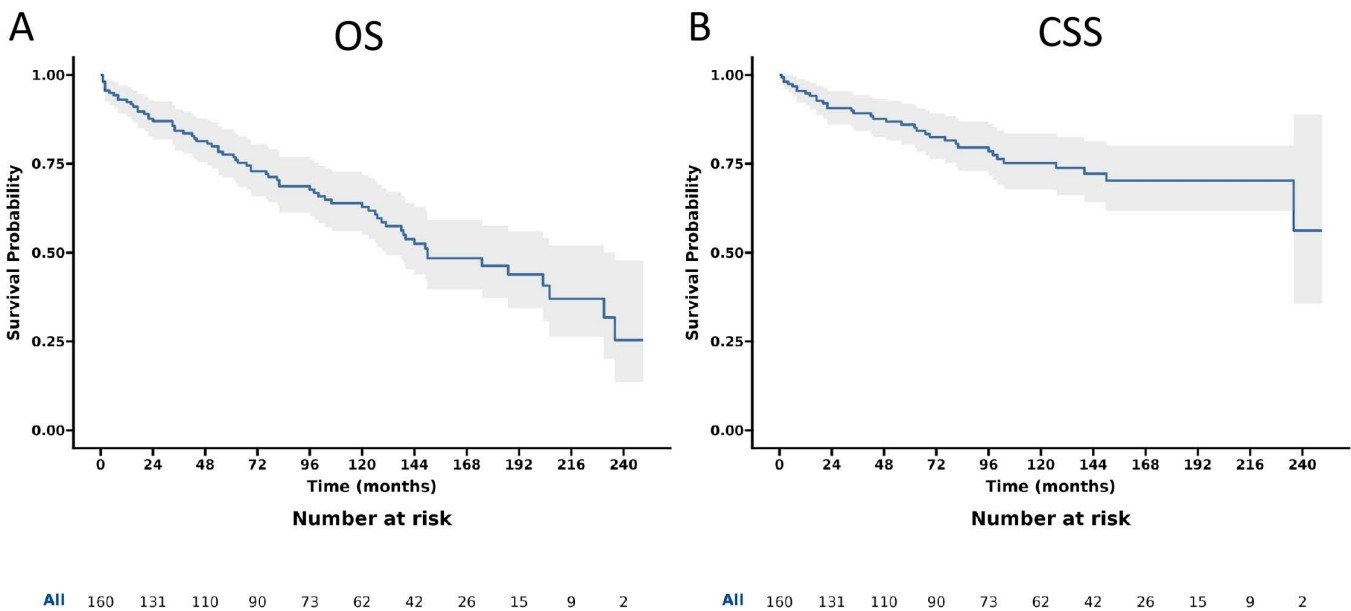

**Fig 2. Overall survival (OS) and cancer-specific survival (CSS) of patients with gastrinoma. A.** OS; B CSS.

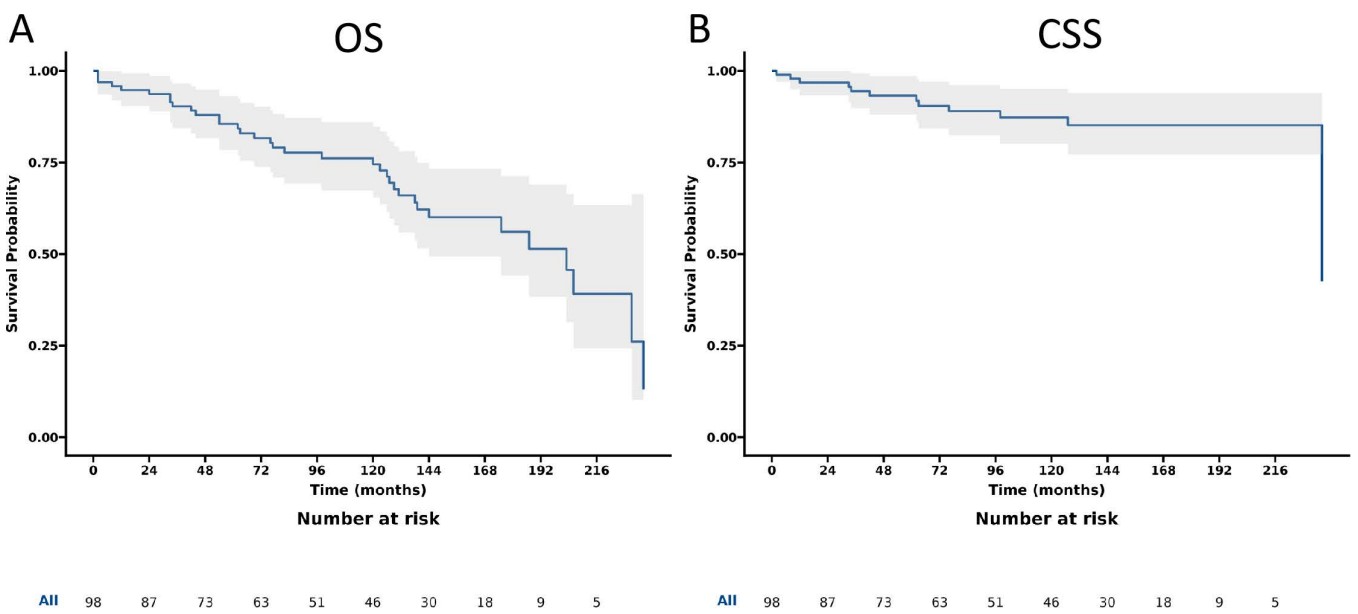

**Fig 3. Overall survival (OS) and cancer-specific survival (CSS) of patients with gastrinoma after surgery. A.** OS; B CSS.

site, gender, T stage, N stage, or lung metastasis (all P > 0.05). However, notable differences emerged in age, tumor grade, and tumor dimensions (all *P* < 0.05). Further multivariate logistic regression analysis identified that being aged 60 years or older (OR = 2.94, P = 0.026) and having a tumor size exceeding 4.0 cm (OR = 6.34, P < 0.001) were independent predictors of liver metastasis, as detailed in Table 6.

**Table 3. Demographic and clinical characteristics of patients with gastrinoma located in the duodenum and pancreas.**

| Variables | Total (n = 136) | Duodenum group (n = 52) | Pancreas group (n = 84) | P-value |
|---|---|---|---|---|
| Marital status, n | | | | 0.803 |
| Married | 74 (54.4%) | 29 (55.8%) | 45 (53.6%) | |
| Unmarried | 62 (45.6%) | 23 (44.2%) | 39 (46.4%) | |
| Age, years, n | | | | 0.747 |
| <60 | 73 (53.7%) | 27 (51.9%) | 46 (54.8%) | |
| ≥60 | 63 (46.3%) | 25 (48.1%) | 38 (45.2%) | |
| Age, years, median (IQR) | 58.0 (47.0, 65.0) | 58.5 (49.3, 65.0) | 57.5 (46.0, 66.5) | 0.745 |
| Race, n | | | | 0.122 |
| White | 105 (77.2%) | 37 (71.2%) | 68 (81.0%) | |
| Black | 23 (16.9%) | 13 (25.0%) | 10 (11.9%) | |
| Others | 8 (5.9%) | 2 (3.8%) | 6 (7.1%) | |
| Grade, n | | | | 0.011 |
| Well | 105 (77.2%) | 45 (86.5%) | 60 (71.4%) | |
| Moderately | 20 (14.7%) | 7 (13.5%) | 13 (15.5%) | |
| Poorly | 11 (8.1%) | 0 (0.0%) | 11 (13.1%) | |
| Sex, n | | | | 0.934 |
| Male | 66 (48.5%) | 25 (48.1%) | 41 (48.8%) | |
| Female | 70 (51.5%) | 27 (51.9%) | 43 (51.2%) | |
| T stage, n | | | | <0.001 |
| T1 | 27 (19.9%) | 11 (21.2%) | 16 (19.0%) | |
| T2 | 41 (30.1%) | 26 (50.0%) | 15 (17.9%) | |
| T3 | 15 (11.0%) | 0 (0.0%) | 15 (17.9%) | |
| T4 | 28 (20.6%) | 0 (0.0%) | 28 (33.3%) | |
| TX | 25 (18.4%) | 15 (28.8%) | 10 (11.9%) | |
| N stage, n | | | | <0.001 |
| N0 | 54 (39.7%) | 27 (51.9%) | 27 (32.1%) | |
| N1 | 66 (48.5%) | 15 (28.8%) | 51 (60.7%) | |
| NX | 16 (11.8%) | 10 (19.2%) | 6 (7.1%) | |
| M stage, n | | | | <0.001 |
| M0 | 105 (77.2%) | 49 (94.2%) | 56 (66.7%) | |
| M1 | 31 (22.8%) | 3 (5.8%) | 28 (33.3%) | |
| Surgery, n | | | | <0.001 |
| Yes | 82 (60.3%) | 42 (80.8%) | 40 (47.6%) | |
| No | 54 (39.7%) | 10 (19.2%) | 44 (52.4%) | |
| Chemotherapy, n | | | | 0.013 |
| No/Unknown | 127 (93.4%) | 52 (100.0%) | 75 (89.3%) | |
| Yes | 9 (6.6%) | 0 (0.0%) | 9 (10.7%) | |
| Tumor size, cm, n | | | | <0.001 |
| ≤2.0 | 68 (50.0%) | 50 (96.2%) | 18 (21.4%) | |
| 2.1-4.0 | 30 (22.1%) | 1 (1.9%) | 29 (34.5%) | |
| >4.0 | 38 (27.9%) | 1 (1.9%) | 37 (44.0%) | |
| Liver metastasis, n | | | | 0.003 |
| No | 111 (81.6%) | 49 (94.2%) | 62 (73.8%) | |
| Yes | 25 (18.4%) | 3 (5.8%) | 22 (26.2%) | |
| Lung metastasis, n | | | | 0.524 |
| No | 134 (98.5%) | 52 (100.0%) | 82 (97.6%) | |

*(Continued)*

**Table 3.** (Continued)

| Variables | Total (n = 136) | Duodenum group (n = 52) | Pancreas group (n = 84) | P-value |
|---|---|---|---|---|
| Yes | 2 (1.5%) | 0 (0.0%) | 2 (2.4%) | |
| Survival months, median (IQR) | 81.0 (34.0, 145.0) | 103.0 (43.0, 154.0) | 69.0 (33.0, 136.0) | 0.093 |

IQR: interquartile range; bold values indicate $P < 0.05$

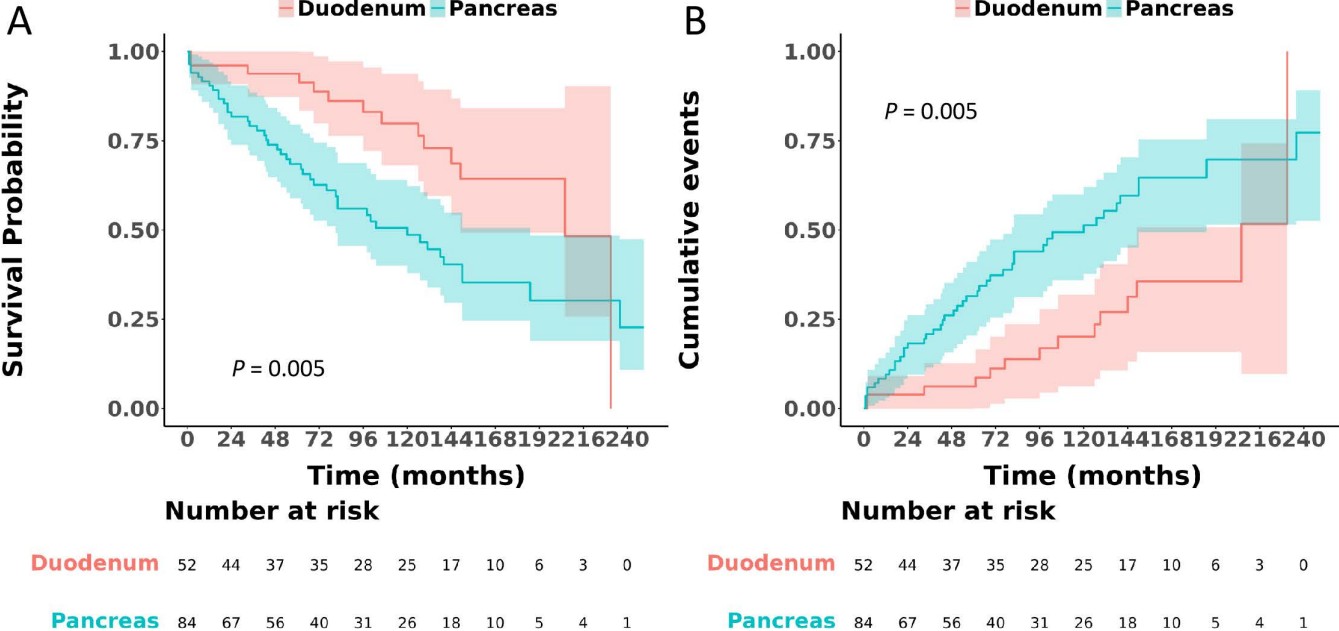

**Fig 4. Comparison of overall survival between patients with gastrinoma located in the duodenum and those in the pancreas.**

## Discussion

In this study, we utilized the SEER database to thoroughly explore the clinicopathological features and survival outcomes of patients with gastrinoma. Our findings demonstrated that gastrinomas originating in the duodenum and pancreas exhibited distinct characteristics; specifically, duodenal gastrinomas generally presented with better cellular differentiation, were diagnosed at an earlier disease stage, and were associated with fewer metastatic events. Furthermore, the prognosis for patients with duodenal gastrinomas proved to be significantly more favorable compared to those with pancreatic gastrinomas, as underscored by higher OS and CSS rates. Through multivariate Cox regression analysis, we identified N stage, surgical intervention, chemotherapy, and tumor size as key risk factors influencing both OS and CSS. Collectively, this research underscores the critical importance of considering tumor location when devising clinical management strategies and assessing prognosis for patients with gastrinoma.

In the study by Falconi et al. [3], it was noted that the age of onset for patients with sporadic gastrinomas ranged from 48 to 55 years, while patients with MEN1 had a significantly younger onset, with ZES manifesting between 32 and 35 years. Norton et al. [22] reported seven cases of primary gastrinomas located in the hepatobiliary tract, with an average age at diagnosis of 43 years. Another study indicated that the mean age at diagnosis for gastrinomas ranged from 20 to 50 years [23]. Huang et al. [24] explored differences between gastric and non-gastric gastrinomas, finding an overall average age of 47 years among 20 patients, with gastric gastrinomas having an average age of 56.3 years, higher

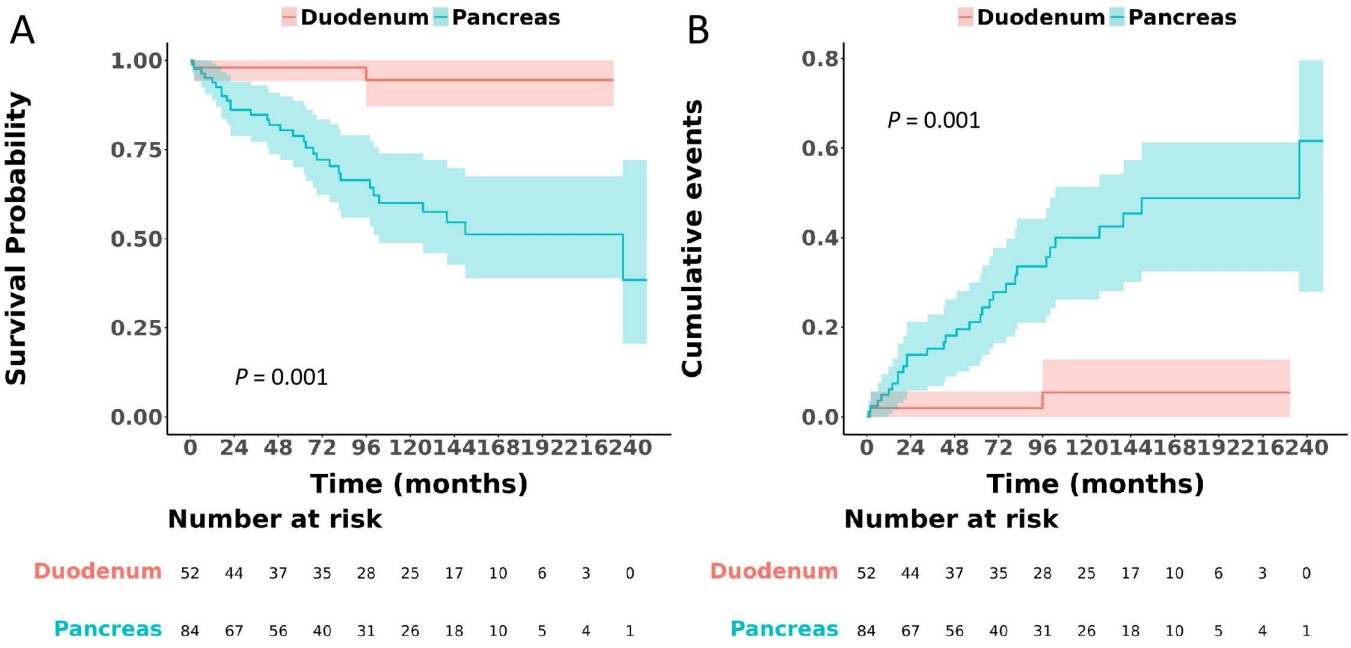

**Fig 5. Comparison of cancer-specific survival between patients with gastrinoma located in the duodenum and those in the pancreas.**

than the 39.7 years observed for non-gastric gastrinomas. In contrast, our study reported a median age of 58.0 years for gastrinomas, slightly higher than previous findings, with the age of onset generally tending to be in middle-aged and younger adult populations. The variations in reported age of onset across studies may stem from differences in sample size, sample origins, and selection criteria; our study, with a relatively larger sample size, offers improved generalizability. Additionally, all participants in our study were recruited from the United States, whereas previous studies included cases from Europe, Taiwan, and other regions. However, advancements in medical technology and revised diagnostic criteria may lead to more accurate and timely diagnoses of gastrinomas, potentially influencing the age at diagnosis. Therefore, future research should aim to collect data from a larger and more diverse patient pool to enhance the representativeness and reliability of the findings. Regarding gender, consistent with previous research, our study found that the incidence rate of gastrinomas is nearly equal between males and females [19,24].

Previous research has revealed that primary gastrinomas can arise from a diverse array of anatomical locations, including the duodenum, pancreas, stomach, mesentery, lymph nodes, heart, lungs, ovaries, hepatobiliary tract, jejunum, and other less common sites [25–32]. Notably, the duodenum and pancreas have consistently been identified as the most frequent sites, accounting for nearly 90% of all gastrinoma cases according to earlier studies [29,30]. In line with these findings, our study also confirms that the duodenum and pancreas are the predominant sites for gastrinoma occurrence. Specifically, among the American population, we observed a higher incidence of pancreatic gastrinomas compared to duodenal gastrinomas. However, the relative incidence of gastrinomas between the duodenum and pancreas remains a subject of debate, highlighting the need for further investigation with larger sample sizes and multi-regional data to draw definitive conclusions. Currently, it is widely acknowledged that gastrinomas are primarily localized within the "Gastrinoma Triangle," which is defined by the junction of the gallbladder and common bile duct superiorly, the junction of the second and third parts of the duodenum inferiorly, and the junction of the neck and body of the pancreas centrally [10,33]. Jensen et al. [34] discovered that the duodenum was the most frequent site for gastrinomas, accounting for roughly 70% of cases. In contrast, our study revealed that the pancreas was the predominant location, with 52.5% of gastrinomas found there,

**Table 4. Univariate and multivariate cox regression for analyzing the overall survival for patients with gastrinoma.**

| Variables | Univariate | | | Multivariate | | |
|---|---|---|---|---|---|---|
| | HR | 95% CI | *P*-value | HR | 95% CI | *P*-value |
| Marital status | | | | | | |
| Married | 1 | Reference | | – | – | – |
| Unmarried | 1.33 | 0.82, 2.15 | 0.244 | – | – | – |
| Age, years | | | | | | |
| <60 | 1 | Reference | | 1 | Reference | |
| ≥60 | 1.65 | 1.02, 2.67 | 0.040 | 1.63 | 1.03, 2.81 | 0.041 |
| Race | | | | | | |
| White | 1 | Reference | | – | – | – |
| Black | 0.71 | 0.36, 1.39 | 0.314 | – | – | – |
| Others | 0.98 | 0.35, 2.73 | 0.975 | – | – | – |
| Location | | | | | | |
| Stomach | 1 | Reference | | – | – | – |
| Duodenum | 0.66 | 0.27, 1.64 | 0.374 | – | – | – |
| Pancreas | 1.58 | 0.71, 3.52 | 0.258 | – | – | – |
| Others | 0.65 | 0.13, 3.14 | 0.592 | – | – | – |
| Grade | | | | | | |
| Well | 1 | Reference | | 1 | Reference | |
| Moderately | 1.29 | 0.55, 3.05 | 0.559 | 1.28 | 0.51, 3.19 | 0.598 |
| Poorly | 5.94 | 3.11, 11.32 | <0.001 | 1.40 | 0.63, 3.13 | 0.408 |
| Sex | | | | | | |
| Male | 1 | Reference | | – | – | – |
| Female | 1.11 | 0.69, 1.79 | 0.662 | – | – | – |
| T stage | | | | | | |
| TX | 1 | Reference | | – | – | – |
| T1 | 1.74 | 0.83, 3.64 | 0.142 | – | – | – |
| T2 | 0.53 | 0.25, 1.14 | 0.106 | – | – | – |
| T3 | 1.02 | 0.40, 2.60 | 0.973 | – | – | – |
| T4 | 1.81 | 0.95, 3.47 | 0.072 | – | – | – |
| N stage | | | | | | |
| N0 | 1 | Reference | | 1 | Reference | |
| N1 | 3.56 | 1.96, 6.48 | <0.001 | 3.96 | 1.85, 8.48 | <0.001 |
| NX | 2.19 | 1.02, 4.68 | 0.044 | 1.40 | 0.61, 3.21 | 0.427 |
| M stage | | | | | | |
| M0 | 1 | Reference | | 1 | Reference | |
| M1 | 2.79 | 1.68, 4.65 | <0.001 | 0.55 | 0.10, 2.98 | 0.487 |
| Surgery | | | | | | |
| Yes | 1 | Reference | | 1 | Reference | |
| No | 1.90 | 1.18, 3.07 | 0.009 | 1.95 | 1.16, 3.30 | 0.012 |

*(Continued)*

**Table 4.** (Continued)

| Variables | Univariate | | | Multivariate | | |
|---|---|---|---|---|---|---|
| | HR | 95% CI | *P*-value | HR | 95% CI | *P*-value |
| Chemotherapy | | | | | | |
| Yes | 1 | Reference | | 1 | Reference | |
| No/Unknown | 4.58 | 2.28, 9.21 | <0.001 | 4.13 | 1.55, 9.33 | 0.001 |
| Tumor size, cm | | | | | | |
| ≤2.0 | 1 | Reference | | 1 | Reference | |
| 2.1-4.0 | 1.11 | 1.03, 1.83 | 0.005 | 1.29 | 1.17, 2.60 | 0.027 |
| >4.0 | 3.13 | 1.89, 5.18 | <0.001 | 2.11 | 1.36, 4.19 | <0.001 |
| Liver metastasis | | | | | | |
| No | 1 | Reference | | 1 | Reference | |
| Yes | 3.67 | 2.17, 6.20 | <0.001 | 1.04 | 0.20, 5.47 | 0.959 |
| Lung metastasis | | | | | | |
| No | 1 | Reference | | – | – | – |
| Yes | 2.91 | 0.40, 21.16 | 0.292 | – | – | – |

HR: hazard ratios; CI: confidence interval; bold values indicate *P*<0.05

while only 32.5% were located in the duodenum. This discrepancy may stem from the fact that Jensen RT's 1999 data were likely influenced by referral biases and the limitations of diagnostic tools available at the time, which led to an overrepresentation of duodenal gastrinoma cases. Our SEER analysis, on the other hand, offers several advantages. It benefits from enhanced pancreatic imaging techniques, a more comprehensive inclusion of non-functional tumors, and population-level sampling. These factors collectively contribute to uncovering a higher true prevalence of pancreatic gastrinomas.

Given their typically slow growth pattern, gastrinomas generally exhibit a favorable prognosis. Our study demonstrated 5- and 10-year OS rates of 77.6% and 62.9%, respectively, along with CSS rates of 86.0% and 75.2%. Notably, for patients undergoing surgical resection, these rates improved significantly, with 5- and 10-year OS reaching 85.5% and 74.5%, and CSS reaching 93.2% and 87.3%, respectively. Consistent with previous reports, surgically treated gastrinomas have been shown to achieve 5- and 10-year survival rates exceeding 80% [35]. However, our study also uncovered potential prognostic variations based on tumor location, revealing that duodenal gastrinomas had a significantly more favorable prognosis compared to pancreatic gastrinomas (P<0.05). This difference may stem from several factors: duodenal tumors are generally more surgically accessible, involve lower operational complexity, and are subject to less compression from adjacent vital organs, thereby reducing surgical risks and postoperative complications; additionally, these tumors may carry a lower risk of distant metastasis, contributing to their improved prognosis. Furthermore, other factors such as tumor differentiation, staging, therapeutic strategies, and individual patient characteristics may also influence prognosis.

Our population-based study reveals several key findings that both corroborate and contrast with prior landmark studies by Yu et al. [36] and Gibril et al. [37]. Regarding age distribution, our SEER data showed peak incidence at 60–69 years, which aligns with Yu et al.'s cohort (median age 50 years at diagnosis) but suggests a later presentation compared to Gibril et al.'s MEN1-associated cases (median age 38 years). This discrepancy likely reflects our inclusion of sporadic gastrinomas (77.5% well-differentiated), whereas Gibril et al.'s cohort predominantly studied MEN1-related tumors known for earlier onset. Tumor location patterns in our study (pancreas 52.5%, duodenum 32.5%) differ notably from both

**Table 5. Univariate and multivariate cox regression for analyzing the cancer-specific survival for patients with gastrinoma.**

| Variables | Univariate | | | Multivariate | | |
|---|---|---|---|---|---|---|
| | HR | 95% CI | *P*-value | HR | 95% CI | *P*-value |
| Marital status | | | | | | |
| Married | 1 | Reference | | – | – | – |
| Unmarried | 1.11 | 0.57, 2.16 | 0.760 | – | – | – |
| Age, years | | | | | | |
| <60 | 1 | Reference | | – | – | – |
| ≥60 | 1.54 | 0.79, 2.99 | 0.206 | – | – | – |
| Race | | | | | | |
| White | 1 | Reference | | 1 | Reference | |
| Black | 0.13 | 0.02, 0.97 | 0.046 | 0.17 | 0.02, 1.83 | 0.144 |
| Others | 1.27 | 0.39, 4.17 | 0.694 | 0.45 | 0.12, 1.61 | 0.218 |
| Location | | | | | | |
| Stomach | 1 | Reference | | – | – | – |
| Duodenum | 0.65 | 0.06, 7.22 | 0.729 | – | – | – |
| Pancreas | 6.89 | 0.94, 50.58 | 0.058 | – | – | – |
| Others | 4.55 | 0.41, 50.20 | 0.217 | – | – | – |
| Grade | | | | | | |
| Well | 1 | Reference | | 1 | Reference | |
| Moderately | 1.92 | 0.21, 3.96 | 0.909 | 1.19 | 0.05, 1.68 | 0.179 |
| Poorly | 15.27 | 7.14, 32.66 | <0.001 | 6.02 | 1.76, 20.57 | 0.004 |
| Sex | | | | | | |
| Male | 1 | Reference | | – | – | – |
| Female | 1.10 | 0.56, 2.13 | 0.786 | – | – | – |
| T stage | | | | | | |
| TX | 1 | Reference | | 1 | Reference | |
| T1 | 1.36 | 0.43, 4.31 | 0.600 | 1.10 | 0.44, 2.82 | 0.144 |
| T2 | 0.12 | 0.01, 0.94 | 0.044 | 0.50 | 0.04, 6.41 | 0.597 |
| T3 | 1.49 | 0.44, 5.11 | 0.524 | 3.00 | 0.46, 19.64 | 0.251 |
| T4 | 3.54 | 1.47, 8.52 | 0.005 | 0.33 | 0.06, 1.75 | 0.192 |
| N stage | | | | | | |
| N0 | 1 | Reference | | 1 | Reference | |
| N1 | 5.90 | 2.23, 15.60 | <0.001 | 4.88 | 1.11, 21.44 | 0.036 |
| NX | 3.24 | 0.99, 10.62 | 0.052 | 2.25 | 0.41, 12.45 | 0.354 |
| M stage | | | | | | |
| M0 | 1 | Reference | | 1 | Reference | |
| M1 | 6.99 | 3.55, 13.77 | <0.001 | 0.70 | 0.05, 9.19 | 0.783 |
| Surgery | | | | | | |
| Yes | 1 | Reference | | 1 | Reference | |
| No | 3.77 | 1.87, 7.61 | <0.001 | 5.87 | 2.18, 15.81 | <0.001 |
| Chemotherapy | | | | | | |
| Yes | 1 | Reference | | 1 | Reference | |
| No/Unknown | 7.64 | 3.48, 16.78 | <0.001 | 4.16 | 0.93, 14.22 | 0.063 |
| Tumor size, cm | | | | | | |
| ≤2.0 | 1 | Reference | | 1 | Reference | |
| 2.1-4.0 | 1.19 | 0.13, 1.98 | 0.231 | 1.54 | 0.08, 4.63 | 0.534 |
| >4.0 | 7.21 | 3.14, 16.59 | <0.001 | 6.10 | 1.12, 33.40 | 0.037 |

*(Continued)*

**Table 5.** (Continued)

| Variables | Univariate | | | Multivariate | | |
|---|---|---|---|---|---|---|
| | HR | 95% CI | *P*-value | HR | 95% CI | *P*-value |
| Liver metastasis | | | | | | |
| No | 1 | Reference | | 1 | Reference | |
| Yes | 9.06 | 4.60, 17.85 | <0.001 | 2.20 | 0.18, 26.91 | 0.537 |

HR: hazard ratios; CI: confidence interval; bold values indicate *P*< 0.05

references. Yu et al. reported duodenal predominance (70%), while Gibril et al. found pancreatic lesions in 55% of sporadic cases. Our higher pancreatic proportion may reflect improved imaging detection of pancreatic tumors in the modern era. Crucially, we confirmed duodenal tumors' superior prognosis (5-year CSS 86% vs pancreatic 75.2%), mirroring Yu et al.'s observation of better survival with duodenal lesions, though their cohort showed lower 10-year survival (51% overall) compared to our 62.9% OS – potentially due to advances in metastatic disease management. Notably, our multivariate analysis identified tumor size and N stage as independent prognostic factors, consistent with Yu et al.'s emphasis on metastatic burden. However, while Yu et al. reported liver metastases as the dominant cause of death (50%), our SEER data showed better metastatic control (5-year CSS 86%), possibly attributable to increased surgical resection rates (49% enucleation/36% pancreatic resection in comparable studies) and somatostatin analogs introduced post-2000.

It is important to note that the associations between surgical intervention, chemotherapy, and survival outcomes should be interpreted with caution. The protective effects observed in our multivariate analysis are likely influenced by confounding by indication. Patients who underwent surgery or received chemotherapy were a selected group; surgical candidates typically had less advanced or resectable disease, while chemotherapy was predominantly administered in cases with more aggressive tumor behavior or metastasis. Therefore, the observed hazard ratios may not represent the true causal effects of these treatments, and prospective studies are needed for definitive evaluation.

Our study has several limitations that should be noted. Firstly, the sample size of 160 gastrinoma patients is relatively small, which may impact the reliability and precision of our findings, particularly evidenced by the wide CIs observed for some HRs, such as the CSS HR of 10.70 (95% CI 2.55–44.90). This limitation underscores the need for caution in interpreting these results and highlights the importance of larger, multi-center studies to validate our findings. However, it is worth mentioning that gastrinomas are rare tumors globally, and our study includes one of the larger populations of gastrinoma patients at the population level to date. Secondly, our research relies on data from the SEER database, which inherently presents risks of data inconsistency and bias. To mitigate this, we utilized MI techniques to handle missing data. Third, the SEER database lacks detailed information on postoperative complications, specific chemotherapy regimens, disease recurrence, and the criteria for selecting surgical interventions for individual patients, which may limit the generalizability of our findings regarding treatment outcomes. Furthermore, as a registry-based study, the SEER database lacks detailed information on clinical presentations, serum gastrin levels, and key pathological markers such as the Ki-67 proliferation index and somatostatin receptor (SSTR) expression status. The inclusion of these clinically significant parameters could provide deeper insights into tumor behavior and prognostication. Lastly, it is important to note that the SEER database primarily reflects the American healthcare system, and our findings may not be directly generalizable to other regions or cultural contexts with differing clinical practices. Therefore, our findings should be interpreted with caution when applied to other geographical or cultural contexts. Despite these limitations, our study is, to our knowledge, the first to report on the clinicopathological characteristics and prognosis of gastrinomas at the population level, and to specifically compare the prognosis of gastrinomas located in different sites.

In conclusion, our study found that among the American population, gastrinomas arise more commonly in the pancreas than in the duodenum, and the liver emerges as the predominant site for metastasis. Through our analysis, age and tumor

**Table 6. Univariate and multivariate analysis of liver metastasis in patients with gastrinoma.**

| Variables | Univariate | | | Multivariate | | |
|---|---|---|---|---|---|---|
| | OR | 95% CI | *P*-value | OR | 95% CI | *P*-value |
| Marital status | | | | | | |
| Married | 1 | Reference | | – | – | – |
| Unmarried | 1.04 | 0.46, 2.32 | 0.926 | – | – | – |
| Age, years | | | | | | |
| <60 | 1 | Reference | | 1 | Reference | |
| ≥60 | 2.47 | 1.06, 5.71 | 0.035 | 2.94 | 1.14, 7.57 | 0.026 |
| Race | | | | | | |
| White | 1 | Reference | | – | – | – |
| Black | 0.35 | 0.08, 1.59 | 0.174 | – | – | – |
| Others | 3.51 | 0.87, 14.11 | 0.077 | – | – | – |
| Location | | | | | | |
| Stomach | 1 | Reference | | – | – | – |
| Duodenum | 0.46 | 0.07, 3.01 | 0.417 | – | – | – |
| Pancreas | 2.66 | 0.56, 12.58 | 0.217 | – | – | – |
| Others | 3.00 | 0.33, 27.23 | 0.329 | – | – | – |
| Grade | | | | | | |
| Well | 1 | Reference | | 1 | Reference | |
| Moderately | 1.25 | 0.03, 1.95 | 0.185 | 1.50 | 0.04, 2.75 | 0.300 |
| Poorly | 6.93 | 2.17, 22.15 | 0.001 | 2.30 | 0.61, 8.68 | 0.218 |
| Sex | | | | | | |
| Male | 1 | Reference | | – | – | – |
| Female | 1.66 | 0.73, 3.79 | 0.227 | – | – | – |
| T stage | | | | | | |
| TX | 1 | Reference | | – | – | – |
| T1 | 0.31 | 0.06, 1.59 | 0.159 | – | – | – |
| T2 | 0.65 | 0.20, 2.12 | 0.474 | – | – | – |
| T3 | 1.11 | 0.25, 5.00 | 0.895 | – | – | – |
| T4 | 2.71 | 0.89, 8.22 | 0.079 | – | – | – |
| N stage | | | | | | |
| N0 | 1 | Reference | | – | – | – |
| N1 | 2.32 | 0.93, 5.80 | 0.072 | – | – | – |
| NX | 1.87 | 0.50, 7.02 | 0.351 | – | – | – |
| Tumor size, cm | | | | | | |
| ≤2.0 | 1 | Reference | | 1 | Reference | |
| 2.1-4.0 | 1.03 | 0.24, 4.37 | 0.969 | 0.94 | 0.22, 4.12 | 0.937 |
| >4.0 | 8.64 | 3.13, 23.84 | <0.001 | 6.34 | 2.03, 19.80 | <0.001 |
| Lung metastasis | | | | | | |
| No | 1 | Reference | | – | – | – |
| Yes | 4.64 | 0.28, 76.48 | 0.283 | – | – | – |

OR: odds ratios; CI: confidence interval; bold values indicate *P*<0.05

size were identified as independent predictors of liver metastasis in gastrinoma patients. Furthermore, we observed significant clinicopathological differences between duodenal and pancreatic gastrinomas, with those located in the duodenum exhibiting a notably more favorable prognosis compared to their pancreatic counterparts.

## Supporting information

**S1 Table. Baseline characteristics of patients with gastrinoma based on the SEER database before multiple imputation.**
(DOCX)

**S1 File. The data.**
(XLSX)

## Author contributions

**Conceptualization:** Feidie Xu.

**Data curation:** Chengjie Lu.

**Formal analysis:** Chenjian Qiu.

**Funding acquisition:** Chengjie Lu, Jingxian Shi, Zhenguo Qiao.

**Investigation:** Jingxian Shi.

**Methodology:** Feidie Xu.

**Software:** Chengru Hu.

**Supervision:** Zhenguo Qiao.

**Validation:** Yaping Wang.

**Writing – original draft:** Chenjian Qiu.

**Writing – review & editing:** Chengru Hu, Zhenguo Qiao, Yaping Wang, Feidie Xu.

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
