## [Decision Letter · Decision Letter 0]

2 Sep 2025

Dear Dr.  Qiao,

Thank you for submitting your manuscript to PLOS ONE. After careful consideration, we feel that it has merit but does not fully meet PLOS ONE’s publication criteria as it currently stands. Therefore, we invite you to submit a revised version of the manuscript that addresses the points raised during the review process.

We look forward to receiving your revised manuscript.

Kind regards,

Yuan-Jia Chen, MD

Academic Editor

PLOS ONE

Journal Requirements:

“This work was supported by the Science and Technology Development Program of Suzhou (SYWD2024077), the Development Fund of Xuzhou Medical University Affiliated Hospital (XYFZ202404 and XYFM202426), the Program for the Talents in Science and Education of Wujiang District, Suzhou, China (Grant No. WWK202117) and the Scientific Research Project of Suzhou Ninth People’s Hospital (YK202117 and YK202438). There was no additional external funding received for this study.”

4. Thank you for uploading your study's underlying data set. Unfortunately, the repository you have noted in your Data Availability statement does not qualify as an acceptable data repository according to PLOS's standards.

5. Please include captions for your Supporting Information files at the end of your manuscript, and update any in-text citations to match accordingly. Please see our Supporting Information guidelines for more information: http://journals.plos.org/plosone/s/supporting-information .

6. Please remove all personal information, ensure that the data shared are in accordance with participant consent, and re-upload a fully anonymized data set.

Additional Editor Comments:

Please revise your MS, according to reviewers’suggestions. Some of important publicatons concerned with gastrinoma should be carefully read before writing your MS. In Jensen's review (R. Jensen Ann. Oncol. 1999; 10: Suppl.4, S170-S176), the most common location of gastrinoma was duodenum (about 70%) but in your MS, most of gastrinomas located in pancreas (52.5%) while only 32.5% of gastrinomas located in duodenum, such differences should be discusse.

Reviewer's Responses to Questions

**Comments to the Author**

1. Is the manuscript technically sound, and do the data support the conclusions?

Reviewer #1: Partly

Reviewer #2: Yes

2. Has the statistical analysis been performed appropriately and rigorously?

Reviewer #1: Yes

Reviewer #2: Yes

3. Have the authors made all data underlying the findings in their manuscript fully available?

Reviewer #1: Yes

Reviewer #2: Yes

4. Is the manuscript presented in an intelligible fashion and written in standard English?

Reviewer #1: Yes

Reviewer #2: Yes

Reviewer #1: This manuscript presents a population-level analysis of gastrinoma patients using the SEER database from 2000 to 2020. The study aims to assess clinicopathological features and survival outcomes, as well as to compare duodenal and pancreatic gastrinomas. Given the rarity of gastrinomas, this research adds valuable epidemiologic insight and provides potentially practice-relevant data regarding prognosis by tumor location.

The study addresses a rare and clinically important topic using a relatively large dataset for such tumors. By comparing duodenal and pancreatic gastrinomas and identifying independent prognostic factors for overall and cancer-specific survival, the manuscript offers novel findings. This stratified analysis is particularly useful for clinical decision-making.

Patient selection via ICD-O-3 coding is appropriate.

The use of multiple imputation (MI) for missing data and Cox regression is sound. However, the rationale behind certain cutoff values (e.g., tumor size at 2.0 cm and 5.0 cm) should be clarified.

While MI is mentioned, the exact method (e.g., number of imputations, convergence diagnostics) should be described in more detail.

The use of both univariate and multivariate Cox and logistic regression models is appropriate.

Kaplan-Meier curves, log-rank tests, and hazard ratios are well presented.

Several hazard ratios have wide confidence intervals, particularly for cancer-specific survival (e.g., HR = 10.70), likely due to small sample sizes. These should be acknowledged and discussed.

Language and Writing Quality is moderate — Revision Required. The authors are strongly encouraged to seek professional English language editing to enhance readability and flow.

Ethical approval is not required for SEER data analysis. This is correctly stated.

There is no identifiable patient data and the authors have declared no competing interests.

Data availability via the SEER database is clearly stated and in accordance with PLOS ONE requirements.

Strengths of the Study:

Use of a population-based dataset (SEER) for a rare tumor.

Comparison of anatomical subtypes with survival implications.

Identification of independent prognostic factors for OS and CSS.

Weaknesses and Suggestions for Improvement

Language issues - Manuscript requires grammatical editing and stylistic refinement.

MI explanation - Add detail regarding multiple imputation methodology.

Clinical limitations - Acknowledge the lack of data on recurrence, chemotherapy type, or surgical details in SEER.

Generalizability - Clearly state that SEER-based findings may not generalize beyond the U.S. system.

Recommendation

Major Revision Required

This manuscript addresses an important and underexplored topic and offers meaningful clinical insights. However, major revisions are necessary, particularly in terms of language quality, data presentation, and methodological clarification.

Reviewer #2: 1. The purpose of this study was to analyze the clinicopathological features and prognosis of patients with gastrinoma using the SEER database. The chi-square test, Kaplan–Meier method, and Cox proportional hazards model were employed for data analysis, yielding valuable insights. For instance, among 160 patients with gastrinoma, prevalence increased with age, peaking in the 60–69 year group. The pancreas was identified as the most common site of occurrence, and pancreatic gastrinomas were associated with a poorer prognosis compared to those originating in the duodenum. The research methodology was sound and effectively addressed the study’s objectives.

2. The availability of the original data could further strengthen the support for the conclusions. Additionally, the inclusion of surgery and chemotherapy in the prognostic analysis may warrant further discussion. For example, some patients missed surgical opportunities due to metastasis, while others did not require chemotherapy after curative resection. Given the complexity of these influencing factors, the results are challenging to interpret, a point that was not adequately addressed in the discussion section.

3. Gastrinomas are rare tumors; however, for a case series spanning 20 years, the number of cases remains relatively small. Larger prospective studies on gastrinoma have already been reported earlier. Moreover, among the 160 gastrinoma cases analyzed in this study, how many were sporadic and how many were MEN-1-related? MEN-1-associated gastrinoma patients typically present at a younger age, and some studies suggest this may correlate with prognosis. If the gastrinoma cases in this study include both sporadic and MEN-1-related types, it is recommended to discuss them separately. As this study and the prior work by Robert T. Jensen et al. both pertain to US patient populations, it would be valuable to explicitly compare the clinical manifestations and prognosis with those findings, and differences should be analized and discussed.

References:

(1) Yu F, et al. J Clin Oncol. 1999;17(2):615-30.

(2) Gibril F, et al. Medicine (Baltimore). 2004;83(1):43-83.

4. Several language issues were noted that affect clarity and precision. For instance, in the “Results — Patient Characteristics” section, the sentence “A study involving 160 patients revealed that the incident rate was followed with age” should replace “incident rate” with “prevalence.” Attention should also be paid to consistency in tense, punctuation, numerical formatting, and terminology throughout the manuscript.

5. Furthermore, improving the clarity of figures is recommended. For example, in Figure 2A and Figure 2B, which depict Overall Survival (OS) and Cancer-Specific Survival (CSS) of gastrinoma patients, respectively, the titles should explicitly indicate “OS” and “CSS” to facilitate reader comprehension.

**Do you want your identity to be public for this peer review?** For information about this choice, including consent withdrawal, please see our Privacy Policy

Reviewer #1: **Yes: ** Nihat Bugra Agaoglu

Reviewer #2: No

---

## [Author Response · Author response to Decision Letter 1]

8 Sep 2025

Dear editor and reviewers

Submission ID PONE-D-25-00053

Title: Clinicopathological Characteristics and Prognosis Analysis of Gastrinoma Based on the SEER Database

By: zhenguo qiao et al

Thank you very much for your letter and for the editors’ and reviewers’ comments concerning our manuscript entitled “Clinicopathological Characteristics and Prognosis Analysis of Gastrinoma Based on the SEER Database” (Submission ID PONE-D-25-00053). These comments are of great reference value to the revision and improvement of our paper and have important guiding significance to our researches. We have studied comments carefully and have made correction. We hope that the revision is acceptable and look forward to hearing from you soon. Revised portion are marked in color in the paper. The main corrections in the paper and the responds to the reviewer’s comments are as flowing:

Journal Requirements:

Response: We have carefully reviewed and ensured that our manuscript and all supporting files now fully comply with PLOS ONE's style and formatting requirements.

“This work was supported by the Science and Technology Development Program of Suzhou (SYWD2024077), the Development Fund of Xuzhou Medical University Affiliated Hospital (XYFZ202404 and XYFM202426), the Program for the Talents in Science and Education of Wujiang District, Suzhou, China (Grant No. WWK202117) and the Scientific Research Project of Suzhou Ninth People’s Hospital (YK202117 and YK202438). There was no additional external funding received for this study.”

Response: This work was supported by the Science and Technology Development Program of Suzhou (SYWD2024077), the Development Fund of Xuzhou Medical University Affiliated Hospital (XYFZ202404 and XYFM202426), the Program for the Talents in Science and Education of Wujiang District, Suzhou, China (Grant No. WWK202117) and the Scientific Research Project of Suzhou Ninth People’s Hospital (YK202117 and YK202438). There was no additional external funding received for this study. The funders had no role in study design, data collection and analysis, decision to publish, or preparation of the manuscript. We include this amended Role of Funder statement in our cover letter.

Response: We have removed all funding-related information from the manuscript as requested.

4. Thank you for uploading your study's underlying data set. Unfortunately, the repository you have noted in your Data Availability statement does not qualify as an acceptable data repository according to PLOS's standards.

Response: We have uploaded our study's complete minimal dataset as Supporting Information files alongside the revised manuscript.

We have updated the Data Availability Statement to: Data Availability Statement: All relevant data are within the paper and its Supporting Information files.

Response: We have now included the captions for all Supporting Information files at the end of the manuscript.

6. Please remove all personal information, ensure that the data shared are in accordance with participant consent, and re-upload a fully anonymized data set.

Response: We have removed all personal information and re-uploaded a fully anonymized dataset.

Response: Thank you for your guidance regarding the reviewer's recommendation to cite specific previously published works. We have carefully reviewed and evaluated the suggested publications, and we confirm that the two additional references requested by Reviewer 2 are highly relevant to our study. As a result, we have incorporated these references into our manuscript to strengthen the context and support our findings. We appreciate the opportunity to enhance the scholarly rigor of our work through this additional citation.

Additional Editor Comments:

Please revise your MS, according to reviewers’suggestions. Some of important publicatons concerned with gastrinoma should be carefully read before writing your MS. In Jensen's review (R. Jensen Ann. Oncol. 1999; 10: Suppl.4, S170-S176), the most common location of gastrinoma was duodenum (about 70%) but in your MS, most of gastrinomas located in pancreas (52.5%) while only 32.5% of gastrinomas located in duodenum, such differences should be discusse.

Response: Thank you for your insightful feedback and for highlighting the discrepancy between our findings and those reported by Jensen et al. We acknowledge the importance of discussing this difference and will revise our manuscript to include a detailed analysis of the potential reasons behind the varying prevalence of gastrinoma locations.

Revised manuscript:

Jensen et al. [34] discovered that the duodenum was the most frequent site for gastrinomas, accounting for roughly 70% of cases. In contrast, our study revealed that the pancreas was the predominant location, with 52.5% of gastrinomas found there, while only 32.5% were located in the duodenum. This discrepancy may stem from the fact that Jensen RT's 1999 data were likely influenced by referral biases and the limitations of diagnostic tools available at the time, which led to an overrepresentation of duodenal gastrinoma cases. Our SEER analysis, on the other hand, offers several advantages. It benefits from enhanced pancreatic imaging techniques, a more comprehensive inclusion of non-functional tumors, and population-level sampling. These factors collectively contribute to uncovering a higher true prevalence of pancreatic gastrinomas.

Reviewer's Responses to Questions

Comments to the Author

Reviewer #1: This manuscript presents a population-level analysis of gastrinoma patients using the SEER database from 2000 to 2020. The study aims to assess clinicopathological features and survival outcomes, as well as to compare duodenal and pancreatic gastrinomas. Given the rarity of gastrinomas, this research adds valuable epidemiologic insight and provides potentially practice-relevant data regarding prognosis by tumor location.

Comment 1

The study addresses a rare and clinically important topic using a relatively large dataset for such tumors. By comparing duodenal and pancreatic gastrinomas and identifying independent prognostic factors for overall and cancer-specific survival, the manuscript offers novel findings. This stratified analysis is particularly useful for clinical decision-making.

Response 1

Thank you for your positive feedback and recognition of the clinical significance and novelty of our study. We greatly appreciate your acknowledgment of the stratified analysis and its potential utility in clinical decision-making. Your comments are highly encouraging, and we are glad that our research contributes valuable insights to the field.

Comment 2

Patient selection via ICD-O-3 coding is appropriate.

Response 2

Thank you for your affirmation regarding our patient selection method using ICD-O-3 coding. We appreciate your recognition of its appropriateness.

Comment 3

The use of multiple imputation (MI) for missing data and Cox regression is sound. However, the rationale behind certain cutoff values (e.g., tumor size at 2.0 cm and 5.0 cm) should be clarified.

Response 3

Thank you for your positive assessment of our use of multiple imputation and Cox regression for handling missing data and analyzing survival outcomes. We appreciate your feedback regarding the need to clarify the rationale behind the cutoff values for tumor sizes. The tumor size cutoffs of 2.0 cm and 5.0 cm were chosen based on a combination of clinical relevance and statistical considerations. These specific thresholds align with commonly used clinical benchmarks in the staging and management of neuroendocrine tumors, including gastrinomas. Additionally, these cutoffs allowed for meaningful stratification of patients into distinct prognostic groups, as evidenced by the significant differences in survival outcomes observed in our analysis. We believe these cutoffs provide a balanced and clinically interpretable approach to categorizing tumor sizes in our study.

Revised manuscript:

Tumor size was grouped into ≤2.0, 2.1-5.0, and >5.0 cm to reflect clinical stages, the risk of metastasis, and prognostic differences.

Comment 4

While MI is mentioned, the exact method (e.g., number of imputations, convergence diagnostics) should be described in more detail.

Response 4

We appreciate the reviewer’s insightful comment regarding the MI method. In response, we have incorporated more detailed information about the MI process into  the “Statistical analysis” section.

Revised manuscript:

Categorical variables were expressed as frequencies and percentages, and Chi-quare tests were employed to evaluate differences among groups. For continuous data that did not follow a normal distribution, the median and interquartile range (IQR) were reported, and comparisons were made using the Mann - Whitney U test. To address missing data in the SEER database, we applied multiple imputation (MI) using the mice package (version 4.1.0) in R. Specifically, we generated 20 imputed datasets by utilizing logistic regression for categorical variables. We assessed convergence both visually through trace plots and statistically using the Gelman-Rubin diagnostic. These assessments confirmed that the imputation models had achieved convergence. The pooled results from these 20 imputed datasets were then used in subsequent analyses to enhance robustness and reduce bias resulting from missing data. For the survival analysis of both OS and CSS, we applied the Kaplan-Meier method. Survival curves were compared using the log-rank test. Additionally, Cox proportional hazards models were used for both univariate and multivariate analyses to calculate hazard ratios (HRs) along with their corresponding 95% confidence intervals (CIs). Variables with a P-value less than 0.05 in the univariate analysis were included in the multivariate Cox regression model. A univariate analysis was carried out to identify potential risk factors associated with liver metastasis. Factors with a P-value below 0.05 from this univariate analysis were subsequently analyzed using a multivariate logistic regression model. All statistical calculations were performed using R software (version 4.1.0). Statistical significance was defined as a P-value less than 0.05.

Comment 5

The use of both univariate and multivariate Cox and logistic regression models is appropriate.

Response 5

Thank you for recognizing the appropriateness of employing both univariate and multivariate Cox and logistic regression models in our study. We appreciate your endorsement of our analytical approach.

Comment 6

Kaplan-Meier curves, log-rank tests, and hazard ratios are well presented.

Response 6

Thank you for acknowledging the clear presentation of Kaplan-Meier curves, log-rank tests, and hazard ratios. We appreciate your positive feedback on our data visualization and analysis.

Comment 7

Several hazard ratios have wide confidence intervals, particularly for cancer-specific survival (e.g., HR = 10.70), likely due to small sample sizes. These should be acknowledged and discussed.

Response 7:

We acknowledge the reviewer’s observation regarding the wide confidence intervals observed for several hazard ratios, particularly in relation to cancer-specific survival (e.g., HR = 10.70). We agree that this is likely attributable to the relatively small sample size in our study. We will address this point in the Discussion section of our paper, explicitly acknowledging the limitations imposed by our sample size and its potential impact on the precision of our hazard ratio estimates.

Revised manuscript:

Firstly, the sample size of 160 gastrinoma patients is relatively small, which may impact the reliability and precision of our findings, particularly evidenced by the wide CIs observed for some HRs, such as the CSS HR of 10.70 (95% CI 2.55-44.90). This limitation underscores the need for caution in interpreting these results and highlights the importance of larger, multi-center studies to validate our findings. However, it is worth mentioning that gastrinomas are rare tumors globally, and our study includes one of the larger populations of gastrinoma patients at the population level to date.

Comment 8

Language and Writing Quality is moderate — Revision Required. The authors are strongly encouraged to seek professional English language editing to enhance readability and flow.

Response 8:

We appreciate the reviewer’s feedback on the language and writing quality of our manuscript. We acknowledge the need for improvement in readability and flow and will seek professional English language editing services to refine the manuscript accordingly. We are committed to ensuring the final version is clear, concise, and accessible to a broad audience. Thank you for your constructive suggestion.

Comment 9

Ethical approval is not required for SEER data analysis. This is correctly stated.

Response 9:

Thank you for acknowledging that ethical approval is not required for SEER data analysis and for confirming our correct statement on this matter. We appreciate your attention to detail.

Comment 10

There is no identifiable patient data and the authors have declared no competing interests.

Response 10:

Thank you for confirming that no identifiable patient data were used and for acknowledging our declaration of no competing interests. We appreciate your thorough review.

Comment 11

Data availability via the SEER database is clearly stated and in accordance with PLOS ONE requirements.

Response 11:

Thank you for recognizing that our data availability statement via the SEER database is clear and meets PLOS ONE’s requirements. We appreciate y

---

## [Decision Letter · Decision Letter 1]

6 Nov 2025

Clinicopathological Characteristics and Prognosis Analysis of Gastrinoma Based on the SEER Database

PLOS ONE

Dear Dr. Qiao,

Thank you for submitting your manuscript to PLOS ONE. After careful consideration, we feel that it has merit but does not fully meet PLOS ONE’s publication criteria as it currently stands. Therefore, we invite you to submit a revised version of the manuscript that addresses the points raised during the review process.

https://journals.plos.org/plosone/s/submission-guidelines#loc-laboratory-protocols . Additionally, PLOS ONE offers an option for publishing peer-reviewed Lab Protocol articles, which describe protocols hosted on protocols.io. Read more information on sharing protocols at https://plos.org/protocols?utm_medium=editorial-email&utm_source=authorletters&utm_campaign=protocols .

We look forward to receiving your revised manuscript.

Kind regards,

Yuan-Jia Chen, MD

Academic Editor

PLOS ONE

Journal Requirements:

Reviewers' comments:

Reviewer's Responses to Questions

**Comments to the Author**

Reviewer #2: All comments have been addressed

Reviewer #3: All comments have been addressed

2. Is the manuscript technically sound, and do the data support the conclusions?

Reviewer #2: Yes

Reviewer #3: Yes

3. Has the statistical analysis been performed appropriately and rigorously?

Reviewer #2: Yes

Reviewer #3: Yes

4. Have the authors made all data underlying the findings in their manuscript fully available?

Reviewer #2: Yes

Reviewer #3: Yes

5. Is the manuscript presented in an intelligible fashion and written in standard English?

Reviewer #2: Yes

Reviewer #3: Yes

Reviewer #2: I thank the authors for their comprehensive response to the previous comments. There are just a few minor points that still require attention.

1. Regarding the stratification of tumor size, the ENET guidelines utilize categories of ≤2 cm, 2-4 cm, and >4 cm. Please consider whether adopting this established criteria would be appropriate for your study.

2. I appreciate the authors' explanation for treatment-related factors in the prognostic analysis. From the presented results, both surgery and chemotherapy appear to be protective factors. However, the discussion states, "Similarly, chemotherapy was predominantly administered in cases with malignant behavior or metastasis, which may explain its association with poorer survival in multivariate analysis." This seems to indicate chemotherapy is associated with poorer survival. Please verify this apparent inconsistency.

3. This study identifies the pancreas as a common site for gastrinomas. Therefore, should the statement, "Our higher pancreatic proportion may reflect improved imaging detection of small duodenal tumors in modern eras," be revised to suggest that it reflects improved detection of pancreatic tumors instead?

Reviewer #3: Gastrinomas are rare tumors classified as a type of neuroendocrine tumor. This study conducted clinicopathological and prognostic analyses based on 20-year case data from the SEER database, featuring a relatively large sample size, comprehensive data collection, and appropriate analytical methods. Ultimately, multivariate analysis revealed that pancreatic-origin gastrinomas have a poor prognosis, and tumor grade, N stage, surgery, chemotherapy, and tumor size are independent risk factors affecting the cancer-specific survival (CSS) of gastrinoma patients, which holds certain reference value for clinical practice.

Due to its reliance on retrospective data from the SEER database, the study lacks data on clinical manifestations, gastrin levels, pathological Ki-67 index, and SSTR2 immunohistochemical results, which may introduce some bias into the findings. However, this is a common limitation of such retrospective studies. If data on this type of tumor from local institutions is combined with SEER database data for further analysis, the research results will be more reference-worthy.

**Do you want your identity to be public for this peer review?** For information about this choice, including consent withdrawal, please see our Privacy Policy

Reviewer #2: No

Reviewer #3: No

---

## [Author Response · Author response to Decision Letter 2]

7 Nov 2025

Dear editor and reviewers

Submission ID PONE-D-25-00053

Title: Clinicopathological Characteristics and Prognosis Analysis of Gastrinoma Based on the SEER Database

By: zhenguo qiao et al

Thank you very much for your letter and for the editors’ and reviewers’ comments concerning our manuscript entitled “Clinicopathological Characteristics and Prognosis Analysis of Gastrinoma Based on the SEER Database” (Submission ID PONE-D-25-00053). These comments are of great reference value to the revision and improvement of our paper and have important guiding significance to our researches. We have studied comments carefully and have made correction. We hope that the revision is acceptable and look forward to hearing from you soon. Revised portion are marked in color in the paper. The main corrections in the paper and the responds to the reviewer’s comments are as flowing:

Reviewer #2: I thank the authors for their comprehensive response to the previous comments. There are just a few minor points that still require attention.

Comment 1

1. Regarding the stratification of tumor size, the ENET guidelines utilize categories of ≤2 cm, 2-4 cm, and >4 cm. Please consider whether adopting this established criteria would be appropriate for your study.

Response 1

We sincerely thank the reviewer for this valuable and insightful suggestion. We agree that aligning our analysis with the established ENETS guideline criteria for tumor size stratification (≤2 cm, 2-4 cm, and >4 cm) will enhance the clinical relevance and interpretability of our findings, and we appreciate the reviewer pointing this out. In accordance with this comment, we have now re-stratified the tumor size variable in our dataset using the recommended categories: ≤2 cm, 2.1-4 cm, and >4 cm. Subsequently, we have repeated all relevant statistical analyses, including: 1) The comparison of clinicopathological characteristics (now reflected in the updated Table 1 and Table 3). 2) The univariate and multivariate Cox regression analyses for Overall Survival (OS) and Cancer-Specific Survival (CSS) (now presented in the revised Table 4 and Table 5). 3) The univariate and multivariate analysis of risk factors for liver metastasis (now shown in the updated Table 6). Thank you once again for this constructive comment, which has undoubtedly improved the quality of our manuscript.

Comment 2

2. I appreciate the authors' explanation for treatment-related factors in the prognostic analysis. From the presented results, both surgery and chemotherapy appear to be protective factors. However, the discussion states, "Similarly, chemotherapy was predominantly administered in cases with malignant behavior or metastasis, which may explain its association with poorer survival in multivariate analysis." This seems to indicate chemotherapy is associated with poorer survival. Please verify this apparent inconsistency.

Response 2

We sincerely thank the reviewer for this exceptionally careful and astute observation regarding the interpretation of the chemotherapy results. The reviewer is absolutely correct to point out the apparent inconsistency between our multivariate analysis results, which show chemotherapy as a protective factor (as indicated by HR > 1 for the "No/Unknown" group in Tables 4 and 5), and our discussion statement, which suggested an association with poorer survival. We apologize for this confusion and the oversight in our manuscript wording. The results presented in Tables 4 and 5 are accurate. In our Cox regression model, the "No/Unknown" chemotherapy group was used as the reference. A Hazard Ratio (HR) greater than 1 for this reference group indeed indicates that receiving chemotherapy ("Yes" group) is associated with a lower risk of death, meaning it appears as a protective factor in the statistical model. However, this statistical finding is almost certainly subject to significant confounding by indication. As we attempted to discuss, patients who received chemotherapy were likely those with more aggressive or advanced disease at baseline. The protective HR likely reflects that, despite having worse prognostic factors, these patients' survival was not as poor as it would have been without chemotherapy, hence the model attributes a "protective" effect to the treatment. Our statement in the discussion that "chemotherapy was predominantly administered in cases with malignant behavior or metastasis, which may explain its association with poorer survival" was incorrectly phrased and created the inconsistency. We intended to explain the underlying clinical context that biases the raw statistical association, but we erroneously used "poorer survival" instead of describing the confounding nature of the indication.

Actions Taken:

We have thoroughly revised the relevant section in the Discussion to remove this contradictory statement and to provide a clearer and more accurate interpretation. The revised text now reads: "It is important to note that the associations between surgical intervention, chemotherapy, and survival outcomes should be interpreted with caution. The protective effects observed in our multivariate analysis are likely influenced by confounding by indication. Patients who underwent surgery or received chemotherapy were a selected group; surgical candidates typically had less advanced or resectable disease, while chemotherapy was predominantly administered in cases with more aggressive tumor behavior or metastasis. Therefore, the observed hazard ratios may not represent the true causal effects of these treatments, and prospective studies are needed for definitive evaluation." Thank you once again for highlighting this critical issue, which has allowed us to improve the clarity and accuracy of our manuscript.

Comment 3

3. This study identifies the pancreas as a common site for gastrinomas. Therefore, should the statement, "Our higher pancreatic proportion may reflect improved imaging detection of small duodenal tumors in modern eras," be revised to suggest that it reflects improved detection of pancreatic tumors instead?

Response 3

We sincerely thank the reviewer for this excellent and logical suggestion. The reviewer is absolutely correct. Our original statement, which suggested that a higher proportion of pancreatic gastrinomas might be due to improved detection of duodenal tumors, was indeed counterintuitive and incorrectly phrased. We fully agree with the reviewer's reasoning. The more plausible explanation, which aligns with our central finding of the pancreas being the most common site, is that advancements in cross-sectional imaging (such as CT and MRI) have particularly enhanced the detection of pancreatic tumors in the modern era.

We have revised the sentence in the Discussion section accordingly. The original sentence: "Our higher pancreatic proportion may reflect improved imaging detection of small duodenal tumors in modern eras." Has been changed to: "Our higher pancreatic proportion may reflect improved imaging detection of pancreatic tumors in the modern era." Thank you again for this valuable correction, which significantly improves the logical consistency of our discussion.

Reviewer #3: Gastrinomas are rare tumors classified as a type of neuroendocrine tumor. This study conducted clinicopathological and prognostic analyses based on 20-year case data from the SEER database, featuring a relatively large sample size, comprehensive data collection, and appropriate analytical methods. Ultimately, multivariate analysis revealed that pancreatic-origin gastrinomas have a poor prognosis, and tumor grade, N stage, surgery, chemotherapy, and tumor size are independent risk factors affecting the cancer-specific survival (CSS) of gastrinoma patients, which holds certain reference value for clinical practice.

Due to its reliance on retrospective data from the SEER database, the study lacks data on clinical manifestations, gastrin levels, pathological Ki-67 index, and SSTR2 immunohistochemical results, which may introduce some bias into the findings. However, this is a common limitation of such retrospective studies. If data on this type of tumor from local institutions is combined with SEER database data for further analysis, the research results will be more reference-worthy.

Response �

We are sincerely grateful to the reviewer for their positive assessment of our work and for providing constructive and insightful suggestions to further strengthen our manuscript. We completely agree with the reviewer that the lack of detailed clinical and pathological data, such as clinical symptoms, serum gastrin levels, Ki-67 index, and SSTR2 expression status, is an important limitation of our study based on the SEER database. As the reviewer rightly pointed out, this is an inherent constraint of large-scale registry data, but incorporating these variables would undoubtedly provide a more comprehensive understanding of gastrinoma. In response to this valuable comment, we have added a statement to the Limitations section of our Discussion to explicitly acknowledge this point. The added text reads: "Furthermore, as a registry-based study, the SEER database lacks detailed information on clinical presentations, serum gastrin levels, and key pathological markers such as the Ki-67 proliferation index and somatostatin receptor (SSTR) expression status. The inclusion of these clinically significant parameters could provide deeper insights into tumor behavior and prognostication." Once again, we thank the reviewer for this excellent suggestion. We believe that addressing this limitation has enhanced the transparency and scholarly rigor of our manuscript. We fully endorse the reviewer's view that future studies combining institutional data with larger databases will be highly valuable.

Yours sincerely,

---

## [Editor Report · Decision Letter 2]

16 Nov 2025

Clinicopathological Characteristics and Prognosis Analysis of Gastrinoma Based on the SEER Database

PONE-D-25-00053R2

Dear Dr. Qiao,

We’re pleased to inform you that your manuscript has been judged scientifically suitable for publication and will be formally accepted for publication once it meets all outstanding technical requirements.

Kind regards,

Yuan-Jia Chen, MD

Academic Editor

PLOS ONE
---

## [Editor Report · Acceptance letter]

PONE-D-25-00053R2

PLOS One

Dear Dr. Qiao,

I'm pleased to inform you that your manuscript has been deemed suitable for publication in PLOS One. Congratulations! Your manuscript is now being handed over to our production team.

Kind regards,

on behalf of

Dr Yuan-Jia Chen

Academic Editor

PLOS One